# Creating synthetic populations in transplantation: A Bayesian approach enabling simulation without registry re-sampling

**Paul R. Gunsalus**[1‡], **Johnie Rose**[2‡*], **Carli J. Lehr**[3], **Maryam Valapour**[3‡], **Jarrod E. Dalton**[1‡]

1 Department of Quantitative Health Sciences, Cleveland Clinic, Cleveland, OH, United States of America, 2 Center for Community Health Integration, Case Western Reserve University, Cleveland, OH, United States of America, 3 Department of Pulmonary Medicine, Cleveland Clinic, Cleveland, OH, United States of America

‡ PRG and JR are contributed equally as first authors. MV and JED are contributed equally as senior authors.
* johnie.rose@case.edu

**Data Availability Statement:** The source code to generate the synthetic populations is available at https://github.com/ClevelandClinicQHS/COMET The SRTR data was obtained through a Data Use

## Abstract

Computer simulation has played a pivotal role in analyzing alternative organ allocation strategies in transplantation. The current approach to producing cohorts of organ donors and candidates for individual-level simulation requires directly re-sampling retrospective data from a transplant registry. This historical data may reflect outmoded policies and practices as well as systemic inequities in candidate listing, limiting contemporary applicability of simulation results. We describe the development of an alternative approach for generating synthetic donors and candidates using hierarchical Bayesian network probability models. We developed two Bayesian networks to model dependencies among 10 donor and 36 candidate characteristics relevant to waitlist survival, donor-candidate matching, and post-transplant survival. We estimated parameters for each model using Scientific Registry of Transplant Recipients (SRTR) data. For 100 donor and 100 candidate synthetic populations generated, proportions for each categorical donor or candidate attribute, respectively, fell within one percentage point of observed values; the interquartile ranges (IQRs) of each continuous variable contained the corresponding SRTR observed median. Comparisons of synthetic to observed stratified distributions demonstrated the ability of the method to capture complex joint variability among multiple characteristics. We also demonstrated how changing two upstream population parameters can exert cascading effects on multiple relevant clinical variables in a synthetic population. Generating synthetic donor and candidate populations in transplant simulation may help overcome critical limitations related to the re-sampling of historical data, allowing developers and decision makers to customize the parameters of these populations to reflect realistic or hypothetical future states.

Agreement (DUA). The authors are thus unable to share this data publicly. However, requests for the SRTR data can be placed via srtr@srtr.org at Scientific Registry of Transplant Recipients.

**Funding:** This project was funded by the Heart, Lung, and Blood Institute of the National Institutes of Health (NHLBI) R01HL153175. Dr. Lehr is supported by NHLBI K08HL159236. The content is solely the responsibility of the authors and does not necessarily represent the official views of the National Institutes of Health. The funding sources had no role in the design or conduct of the study; collection, management, analyses, or interpretation of the data; preparation, review, or approval of the manuscript; or decision to submit the manuscript for publication. The funding sources did not provide input or contribute to the analysis or conclusions of this manuscript.

**Competing interests:** The authors have declared that no competing interests exist.

**Abbreviations:** COMET, Computational Open-source Model for Evaluating Transplantation; CAS, Continuous Allocation Score; DAGs, directed acyclic graphs; DCD, donation after cardiac death; ECMO, extracorporeal membrane oxygenation; $FEV_1$, forced expiratory volume in 1 second; FVC, forced vital capacity; $FiO_2$, fraction inspired oxygen; HRSA, Health Resources and Services Administration; IQRs, interquartile ranges; (LAS) (1, 2) (1, 2), Lung Allocation Score; NASEM, National Academies of Science, Engineering, and Medicine; OPTN, Organ Procurement and Transplantation Network; $PaO_2$, partial pressure of arterial oxygen; $PCO_2$, partial pressure of carbon dioxide; P/F ratio, ratio of $PaO_2$ to $FiO_2$; SRTR, Scientific Registry of Transplant Recipients; SAMs, Simulated Allocation Models; TSAM, Thoracic Simulated Allocation Model.

# Introduction

The Simulated Allocation Models (SAMs) developed by the U.S. Scientific Registry of Transplant Recipients (SRTR) have played essential roles in forecasting the impact of future policies on patient populations prior to implementation [1–5]. Existing SAMs include the Liver Simulated Allocation Model [6], the Kidney/Pancreas Simulated Allocation Model [7], and the Thoracic Simulated Allocation Model (TSAM) [8].

Each of these models generates candidate and donor populations by re-sampling the profiles of historical donors and candidates whose data were captured in the SRTR. This approach is straightforward but has important implications for the utility of the SAMs in examining future states. First, donor and candidate populations cannot be changed in response to contemporary or hypothetical clinical practice advances, policy changes, demographic shifts, or epidemiologic trends. For example, advances in *ex vivo* organ perfusion [9–11], new therapeutic treatments of underlying diseases [12–16], and broadening transplant eligibility criteria [17–20] likely render historical data far less useful as a predictor of future donor and candidate population profiles. This problem has been illustrated concretely by inaccuracies in model projections attributed to rapid changes in candidate and donor populations [21–23].

Second, biases in the processes that determine who becomes a transplant candidate may artificially lower the number of candidates from traditionally underserved groups [24–26]. Re-sampling SRTR data to build simulated candidate populations therefore runs the risk of propagating these biases in simulation results that could inform real-world policies. Without correcting for these biases in future simulation studies, efforts to achieve the goal of increasing equity in organ allocation algorithms—long pursued by the transplant community and codified by the National Academies of Science, Engineering, and Medicine (NASEM) 2022 report *Realizing the Promise of Equity in the Organ Transplantation System* [26]—could not only fall short, but could in fact yield unintended negative consequences.

Bayesian networks are probabilistic models in which the conditionalities among a set of variables are represented via a graphical model. They can be applied to make relationships among donor or candidate characteristics explicit in a manner that is informed by the underlying SRTR historical data, but that allows manipulation of population parameters to shape the jointly varying characteristics of simulated populations. In this study, we developed and applied Bayesian networks to create synthetic U.S. lung transplant donor and candidate populations and assign realistic characteristics to these synthetic donors and candidates. As an initial validation, we evaluated the ability of these models to faithfully reproduce the characteristics of candidates and donors observed in the SRTR. We also demonstrated how changing two upstream parameters to reflect ongoing trends in candidate populations can exert cascading effects on multiple relevant clinical variables in a synthetic population of candidates.

# Methods

## Model specification and estimation

Our procedure for creating populations of synthetic candidates and donors was implemented in two steps. First, we developed a model to estimate the annual rates at which new donors emanate from each hospital in the U.S. and at which new candidates are listed at each transplant center. Second, we developed donor- and candidate-specific Bayesian network models capable of generating realistic combinations of relevant traits for these respective populations.

**Step 1: Generation of site-specific donor and candidate counts.** In this step, we obtained representative annual counts of donors from each hospital and of adult candidates from each

transplant center in the U.S. We estimated hierarchical Bayesian Poisson models, clustering observations at the hospital or center level for donors and candidates, respectively. The response variable for each model was the number of donors (or candidates) arising from a given site over the length of an observation period for that site. This study used data from the Scientific Registry of Transplant Recipients (SRTR). The SRTR data system includes data on all donor, waitlisted candidates, and transplant recipients in the U.S., submitted by the members of the Organ Procurement and Transplantation Network (OPTN). The Health Resources and Services Administration (HRSA), U.S. Department of Health and Human Services provides oversight of the activities of the OPTN and SRTR contractors.

We extracted data on 15,826 U.S. lung donors between January 1, 2015, and June 30, 2021, whose organs were either utilized for transplant or recovered but not used. Based on an analysis of the donor data, we assumed that 94% of organs recovered for transplant would be transplanted. We included donors who were 12 years of age and older since the majority of adolescent donor organs go to adult candidates. We estimated separate donor rates for each six-month period to account for fluctuations in donor volume. Donors originated from one of 1,691 U.S. hospitals. Since many of these hospitals were of low volume, we only assigned a hospital its own unique set of rate parameters when the hospital had more than two donors between 2013 and 2014. All hospitals with two or fewer donors were assigned a shared set of rate parameters; this latter measure was taken to ensure stability of hospital-level donor recovery rate estimates.

To assign individual U.S. transplant centers their own parameters representing the rates at which they listed new lung transplant candidates, we used SRTR data for 18,299 candidates who were placed on the waiting list from February 19, 2015 (the effective date of a Lung Allocation Score [LAS] organ allocation algorithm update [27]) to September 1, 2021. We included only candidates aged 18 and older since organ allocation rules differ for pediatric candidates [28]; and we excluded candidates for whom there were missing data for the key variables of forced expiratory volume in 1 second ($FEV_1$), forced vital capacity (FVC), partial pressure of carbon dioxide ($PCO_2$), partial pressure of arterial oxygen ($PaO_2$), P/F ratio (ratio of $PaO_2$ to fraction inspired oxygen [$FiO_2$]), and mean pulmonary artery pressure. We also excluded from the analysis centers listing fewer than 10 candidates between 2015 and 2021 and those listing no candidates between 2020 and 2021. We estimated separate candidate rate parameters for each of the remaining 61 transplant center for each diagnosis group used in the U.S. transplant system (Group A, obstructive lung disease; Group B, pulmonary vascular disease; Group C, cystic fibrosis and immunodeficiency disorders; Group D, restrictive lung diseases [29]; and we created separate estimates for each one-month period to account for fluctuations in center volume.

**Step 2: Assignment of donor and candidate characteristics.** We employed hierarchical Bayesian network models (See description and illustration of the basic properties of hierarchical Bayesian network models in **Fig A in S1 Text**) to separately estimate sets of conditional probability distributions for relevant donor and candidate characteristics, respectively. The specific characteristics to be modeled were selected based on their use in either screening candidates for eligibility for a particular donor lung [28]; or for donor-candidate matching, calculating waitlist survival, or calculating post-transplant survival according to either the Lung Allocation Score (LAS) [27, 30] or the more recent Composite Allocation Score (CAS) allocation algorithms used in the U.S. transplantation system [31]. The U.S. lung allocation system and its risk models have served as the template for numerous other transplant systems internationally [32–34]. In 2021, in fact, the LAS model or some adaptation thereof governed greater than 50% of lung transplant activity worldwide [35, 36].

**Table 1. Variable relationships for donor characteristics.**

| Outcome/Descendent Variable | Predictor(s)/Parent Variable(s) | Distribution | Transformations/Other Notes |
|---|---|---|---|
| Hospital/Date | | Poisson | Donors are randomly generated over time by sampling from hospital-specific Poisson distributions. |
| Sex | Hospital | Binomial | Separate distribution for each hospital |
| Cause of Death | Hospital | Multinomial | Separate distribution for each hospital |
| Height | Sex | Normal | |
| DCD status | Cause of Death | Binomial | |
| Age | Cause of Death | Skew Normal | |
| Race/Ethnicity | Hospital | Multinomial | |
| Blood Type | Race/Ethnicity | Multinomial | |
| >20 pack year Smoking History | Age, Sex, Race/Ethnicity | Binomial | |
| Lungs Available (Double, Left, or Right) | — | Multinomial | |

Based on lung donor data from the Scientific Registry of Transplant Recipients (SRTR) between January 1, 2015 and June 30, 2021. Each variable is a node of a Bayesian network, in which parent variables act as predictors for their linked outcome variable(s). For a table of final priors please see Table A in S1 Text. DCD = donation after cardiac death

The structures of the donor and candidate network models are too large to present clearly in visual form; rather, we present the composition of the networks, including dependencies among the included variables and modeled probability distributions for each variable, in tabular format. See **Table 1** for the network of donor characteristics and **Table 2** for the network of candidate characteristics. The dependencies represented by the network structures were based on known clinical and physiologic relationships and confirmed empirically within the SRTR data.

In our network of candidate risk factors, we included summary indices of airway function and of oxygen function, as well as a categorical variable representing the predicted requirement for respiratory support. We have described the estimation of these latent constructs previously [37]: 1) an Airway Function Index representing shared variation among $FEV_1$, FVC and $PCO_2$; 2) an Oxygen Function Index representing shared variation among P/F Ratio, $PaO_2$, and mean pulmonary artery pressure; and 3) a Respiratory Support Cluster variable representing groups defined by oxygen frequency, supplemental oxygen requirement, ventilator status and extracorporeal membrane oxygenation (ECMO). Use of these three existing constructs as variables within our candidate risk factor network provided an efficient means to ensure that the underlying indicators defining each respective construct were not simulated independently from one another, but rather in a manner that preserved covariance among the underlying indicators. Airway Function Index and Oxygen Function Index were used as parent variables (predictors) for the distribution of the Respiratory Support Cluster, which had the following four categories: (1) As needed oxygen ($O_2$), (2) Continuous $O_2$/No positive pressure ventilation (PPV, defined as continuous positive airway pressure, inspiratory positive airway pressure, and invasive mechanical ventilation), (3) Continuous $O_2$/PPV, and (4) Continuous $O_2$/PPV + ECMO.

Distributions of candidate sex, race, and diagnosis groups were estimated separately for each center. We used the Stan probabilistic programing language, applying the No U-turn Sampler (NUTS) method, via the *rstan* R package version 2.21.8, to estimate models for all variables in the donor and candidate networks. We estimated all models using R version 4.2.3 and RStudio Server Edition, Version 2022.07.01 (Posit Software, PBC, Boston, MA) installed on a

**Table 2. Variable relationships for candidate risk factors.**

| Outcome/Descendent Variable | Predictor(s)/Parent Variable(s) | Distribution | Transformations/Other Notes |
|---|---|---|---|
| Center/Date/Diagnosis Group | | Poisson | Candidates are randomly generated over time by sampling from hospital-specific Poisson distributions by Diagnosis Group. |
| Sex | Diagnosis Group | Binomial | |
| Race/Ethnicity | Center | Multinomial | |
| Height | Sex, Diagnosis Group | Normal | |
| Weight | Sex, Diagnosis Group, Height | Normal | Omitted Group C (See below) |
| Weight (Group C) | Sex, Height | Skew Normal | Excluded top and bottom 1% |
| Age | Diagnosis Group | Skew Normal | |
| Airway Function Index | Sex, Diagnosis Group | Mixed Effects (Skew Normal) | Excluded top 1% for Diagnosis Group A |
| Oxygen Function Index | Sex, Diagnosis Group, Airway Function Index | Student's T-Distribution | |
| Respiratory Support Cluster* | Sex, Diagnosis Group, Age, Airway Function Index, Oxygen Function Index | Multinomial | |
| $FEV_1$ | Airway Function Index, Sex, Diagnosis Group, Age | Normal | Excluded top 2% for Diagnosis Groups A and C |
| FVC | Airway Function Index, Sex, Diagnosis Group, Age | Normal | |
| $pCO_2$ | Airway Function Index, Sex, Diagnosis Group | Normal | $1/x$ transformed |
| P/F Ratio | Oxygen Function Index, Sex, Diagnosis Group, Age | Normal | square-root transformed |
| $pO_2$ | Oxygen Function Index, Sex, Diagnosis Group | Normal | $1/x$ transformed |
| Mean Pulmonary Artery Pressure (PAP) | Oxygen Function Index, Sex, Diagnosis Group, Age | Normal | $Ln(x)$ transformed |
| Oxygen Frequency | Sex, Diagnosis Group, Respiratory Support Cluster, Age, Airway Function Index, Oxygen Function Index | Multinomial | |
| Ventilator | Sex, Diagnosis Group, Respiratory Support Cluster, Age, Airway Function Index, Oxygen Function Index | Multinomial | |
| Supplemental Oxygen Requirement | Sex, Diagnosis Group, Respiratory Support Cluster, Age, Airway Function Index, Oxygen Function Index | Ordered logistic regression | By respiratory support group |
| Six Minute Walk Distance | Sex, Diagnosis Group, Respiratory Support Cluster, Age, Airway Function Index, Oxygen Function Index | Zero Inflated Skew Normal | Zero inflation by respiratory support and diagnosis group |
| Bilirubin | Diagnosis Group, Age, Sex | Normal | $Ln(x)$ transformed |
| Creatinine | Diagnosis Group, Age, Sex | Normal | $Ln(x)$ transformed |
| Systolic PAP | Mean Pulmonary Artery Pressure, Diagnosis Group, Age | Normal | Excluded the top 3% within each diagnosis group, Group B has a separate standard error estimation |
| Cardiac Index | Diagnosis Group, Height, Weight | Normal | $Ln(x)$ transformed |
| Central Venous Pressure | Mean Pulmonary Artery Pressure, Diagnosis Group | Gamma | Separate shape and scale parameters by diagnosis group |
| Functional Status | Respiratory Support Cluster, Diagnosis Group | Multinomial | |
| $pCO_2$ Threshold | Respiratory Support, Diagnosis Group, $pCO_2$ | Binomial | $PCO_2 > 45$ mmHg |
| Blood Type | Race/Ethnicity | Multinomial | |

(*Continued*)

**Table 2.** (Continued)

| Outcome/Descendent Variable | Predictor(s)/Parent Variable(s) | Distribution | Transformations/Other Notes |
|---|---|---|---|
| Surgical Type | Diagnosis Group | Multinomial | Double-, single- or either-lung transplants.<br>Single- is calculcated based on left only or right only lung preference. Double- is both only lung preference, Either- is both lung and right/left preference or all lungs preference |
| Diabetes | Sex, Age, Race Ethnicity | Binomial | |
| Sarcoidosis | Center | Binomial | Group A with missing/<30 mean PAP;<br>Group D with >30 mmHg mean PAP |
| Eisenmenger Syndrome | Intercept Only | Binomial | Group B Only |
| Lymphangioleiomyomatosis (LAM) | Intercept Only | Binomial | Group A females, not diagnosed with sarcoidosis |
| Other Specific Diagnoses from Lung Allocation Score models<br>• Bronchiectasis<br>• Pulmonary Fibrosis (other)<br>• Constrictive Bronchiolitis<br>• Bronchiolitis Obliterans | Center | Multinomial | Group A/D not diagnosed with sarcoidosis. Only generated those diagnoses considered for LAS. Otherwise coded as "other diagnosis". COVID-19 fibrosis not simulated, as it was a pandemic related event it will be assumed pulmonary fibrosis (other). |

*(1) As needed oxygen (O2), (2) Continuous O2/No positive pressure ventilation (PPV, defined as continuous positive airway pressure, inspiratory positive airway pressure, and invasive mechanical ventilation), (3) Continuous O2/PPV, and (4) Continuous O2/PPV + ECMO.

Based on lung transplant candidate data from the Scientific Registry of Transplant Recipients (SRTR) between February 19, 2015 and September 1, 2021. Each variable is a node of a Bayesian network, in which parent variables act as predictors for their linked outcome variable(s). For a table of final priors please see Table B in S1 Text.

$FEV_1$ = Forced Expiratory Volume in 1 second; FVC = Forced Vital Capacity (% predicted); PCO2 = partial pressure of carbon dioxide; P/F ratio = ratio of partial pressure of arterial oxygen (PaO2) to fraction inspired oxygen (FiPO2); PO2 = partial pressure of oxygen; Mean PAP = mean pulmonary arterial pressure; Systolic PAP = systolic pulmonary arterial pressure; CVP = central venous pressure. Diagnosis Groups = Group A, obstructive lung disease; Group B, pulmonary vascular disease; Group C, cystic fibrosis and immunodeficiency disorders; and Group D, restrictive lung diseases

Unix server located within the Cleveland Clinic Lerner Research Institute's local computing environment.

## Model validation

The resulting posterior probability distributions were sampled to create 100 synthetic populations of donors and 100 synthetic populations of candidates corresponding to the time periods reflected in the original SRTR cohorts. We compared distributions of the numbers of synthetic candidates generated for each center over 100 runs to actual site-specific numbers reflected in the SRTR data. We compared synthetic donor and candidate characteristics against actual observations in the SRTR data over the same period, using univariable summary statistics.

**Example use case.** To demonstrate how our approach might be applied to generate more realistic future populations for policy simulations, we used the candidate generation model to synthesize a population with diagnosis group and age structure we might expect to observe between January 1, 2021 and July 15, 2027 (an interval equal in length to the original February 19, 2015 through September 1, 2021 model training interval). To do so, we extrapolated annual trends in the distribution of diagnosis groups A through D, and in the diagnosis group-specific mean candidate ages observed from 2015 through the end of 2020 (the last year for which full-year SRTR data was available at the time of writing). This resulted in a hypothesized population from 2021 to 2027 with 17% fewer diagnosis group A candidates, 56% more group B candidates, 70% fewer group C candidates, and 14% more Group D candidates on average—compared to 2015–2020. This large shift partly stems from the introduction and rapid uptake of transmembrane conductance regulator modulator therapies, a drug class which has dramatically improved lung function for many Cystic Fibrosis (diagnosis group C) patients [12]. The

mean ages of groups A through D were projected to increase by 1.5, 4.6, 0.4, and 1.1 years, respectively. We examined the cascading effects of these population changes on the distributions of other candidate characteristics by comparing the combined results from 100 simulations under this forecasted scenario with 100 simulations under the base case (2015–2020) scenario.

The study was approved as exempt research by the Cleveland Clinic Institutional Review Board (20–373) and performed under a data use agreement with the SRTR. Informed consent was waived due to the presence of minimal risk and deidentified data. The source code to generate the synthetic populations is available at https://github.com/ClevelandClinicQHS/COMET.

## Results

The sizes of the 100 distinct synthetic lung donor populations ranged from 15,470 to 16,209 individuals, with a median of 15,854; in the SRTR data, there were 15,826 donors. The sizes of the 100 synthetic candidate populations ranged from 17,849 to 18,741, with a median of 18,284 individuals; in the SRTR data, there were 18,299 candidates. For each of the 61 U.S. transplant centers modeled, **Fig 1** compares the distribution of candidates from each of 100 synthetic candidate populations to the corresponding observed numbers. Only one center had more than two single synthetic populations with an outlying synthetic candidate count (defined as +/-1.5*interquartile range [IQR] and represented as black dots in Fig 1). For each center, the IQR of synthetic candidate counts contained the observed count.

**Tables 3** and **4** compare the characteristics of SRTR donors (Table 3) and candidates (Table 4) to those of simulated donors and candidates, respectively. For each generated donor or candidate trait, the tables show the characteristics of five individual synthetic populations—corresponding to the populations having the minimum, 25th percentile, median, 75th percentile, and maximum sizes. The final column of the table shows the characteristics of all 100 synthetic populations combined. Across individuals from all 100 synthetic donor populations, the proportions describing categorical donor characteristics consistently fell within one percentage point of observed. The median height of synthetic donors was slightly but systematically lower than that for actual donors, with synthetic donors having a median height of 171 cm, while median observed donor height was 173 cm. Fifth, 25th, 75th, and 95th percentile values of height in synthetic and observed donors, however, were closely aligned: 155, 164, 179, and 188 versus 155, 165, 178, and 188, respectively. For the 100 synthetic donor populations combined, the IQRs of both continuous donor variables (height and age) contained the corresponding SRTR observed median.

Across individuals from all 100 synthetic candidate populations, the proportions describing categorical candidate characteristics consistently fell within one percentage point of observed proportions. The candidate model generated slight but systematic over- or underestimates for a subset of continuous clinical variables: Airway Function Index, FVC, P/F ratio, PaO$_2$, six-minute walk distance, bilirubin, and cardiac index. Among the most substantial of these differences were those for FVC (49 [5th/25th/75th/95th percentiles: 22, 37, 61, 80] synthetic versus 47 [5th/25th/75th/95th percentiles: 25, 37, 60, 82] observed) and P/F ratio (275 [5th/25th/75th/95th percentiles: 131, 222, 333, 478] synthetic versus 281 [5th/25th/75th/95th percentiles: 106, 224, 343, 494] observed). Those two variables had non-zero prevalence of missing values in the SRTR data, which is suggestive of non-random missingness as a potential contributing factor for the observed differences. Nonetheless, for the 100 synthetic candidate populations combined, the IQRs of every continuous candidate variable contained the corresponding SRTR observed median. For the majority of the continuous variables the tail behavior (5th and 95th percentiles) closely reflected the tail behavior of the original data.

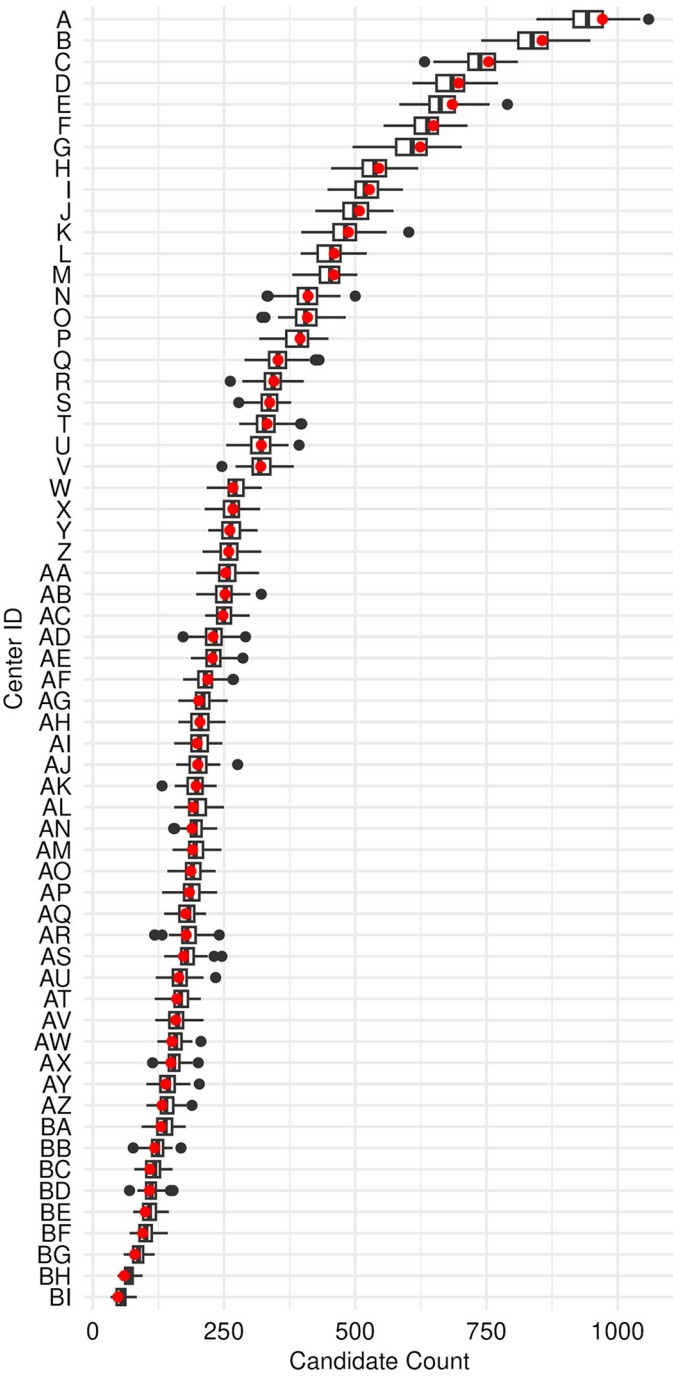

**Fig 1. Box-and-whisker-plots comparing the distribution of synthetic candidates (from 100 synthetic populations) across transplant centers to the corresponding center-specific counts observed in SRTR data (represented by red dots) covering the period February 19, 2015 to September 1, 2021.** Each box horizontally spans the interquartile range (IQR), while the vertical line inside each box represents the median. The extremes of each whisker mark the observation that is farthest to the left or right of the median while still falling within 1.5 times the IQR. Black dots represent outlying synthetic population counts, defined as greater than +/-1.5 times the IQR from the third and first quartiles respectively.

**Table 3. Comparison of observed SRTR and 100 synthetically generated donor populations.**

| Characteristic | Observed | Synthetic Minimum Count | Synthetic First Quantile Count | Synthetic Median Count | Synthetic Third Quantile Count | Synthetic Maximum Count | 100 Synthetic Runs |
|---|---|---|---|---|---|---|---|
| | N = 15,826* | N = 15,470* | N = 15,746* | N = 15,854* | N = 15,949* | N = 16,209* | N = 1,584,991* |
| **Sex** | | | | | | | |
| Female | 60.3% | 60.7% | 60.2% | 60.6% | 60.7% | 59.7% | 60.4% |
| Male | 39.7% | 39.3% | 39.8% | 39.4% | 39.3% | 40.3% | 39.6% |
| **Race/Ethnicity** | | | | | | | |
| NH White | 61.1% | 60.5% | 59.7% | 60.9% | 61.4% | 60.4% | 60.7% |
| NH Black | 18.3% | 18.6% | 18.2% | 18.7% | 18.4% | 18.3% | 18.4% |
| Hispanic | 16.1% | 16.2% | 16.7% | 16.1% | 15.8% | 16.5% | 16.1% |
| Asian | 3.2% | 3.4% | 3.7% | 2.9% | 3.3% | 3.3% | 3.3% |
| AI/AN | 0.6% | 0.7% | 0.7% | 0.6% | 0.5% | 0.7% | 0.7% |
| Pacific Islander | 0.2% | 0.4% | 0.3% | 0.4% | 0.2% | 0.4% | 0.3% |
| Multi/Other | 0.3% | 0.3% | 0.6% | 0.4% | 0.3% | 0.3% | 0.4% |
| **Height (cm)** | 173 [165, 178]; (155, 188) | 171 [164, 179]; (155, 188) | 171 [164, 179]; (154, 188) | 171 [164, 179]; (155, 188) | 171 [164, 179]; (155, 188) | 171 [164, 179]; (154, 188) | 171 [164, 179]; (155, 188) |
| **Cause of Death** | | | | | | | |
| Anoxia | 31.8% | 31.0% | 31.6% | 30.6% | 31.1% | 31.0% | 31.2% |
| Cerebro-vascular | 27.0% | 26.6% | 26.5% | 26.8% | 26.9% | 27.9% | 27.0% |
| Head Trauma | 38.7% | 39.5% | 39.1% | 40.0% | 39.3% | 38.6% | 39.1% |
| CNS Tumor | 0.5% | 0.7% | 0.5% | 0.5% | 0.6% | 0.6% | 0.6% |
| Other | 2.1% | 2.3% | 2.3% | 2.1% | 2.1% | 2.0% | 2.1% |
| **Blood Type** | | | | | | | |
| A | 36.0% | 35.9% | 35.4% | 36.2% | 36.3% | 36.4% | 36.1% |
| AB | 2.1% | 2.1% | 2.0% | 2.3% | 2.0% | 2.2% | 2.1% |
| B | 11.0% | 11.4% | 10.9% | 10.8% | 11.1% | 11.4% | 11.1% |
| O | 50.8% | 50.6% | 51.7% | 50.8% | 50.6% | 50.0% | 50.8% |
| **Age (years)** | 33 [24, 46]; (16, 59) | 33 [24, 44]; (16, 58) | 33 [24, 45]; (16, 59) | 33 [24, 45]; (16, 58) | 33 [24, 45]; (16, 59) | 34 [25, 44]; (16, 59) | 33 [24, 45]; (16, 59) |
| **Donor Organs** | | | | | | | |
| Double Lung | 90.2% | 89.3% | 90.6% | 90.2% | 90.3% | 90.4% | 90.2% |
| Left Lung | 5.7% | 6.1% | 5.5% | 5.5% | 5.7% | 5.3% | 5.7% |
| Right Lung | 4.1% | 4.6% | 3.9% | 4.3% | 4.0% | 4.4% | 4.1% |
| **>20 Pack Years** | 7.5% | 7.8% | 7.5% | 7.7% | 7.5% | 8.2% | 7.8% |
| **DCD** | 6.0% | 5.8% | 5.8% | 5.8% | 6.3% | 5.9% | 6.0% |

* %; Median [IQR]; (5th percentile, 95th percentile)

The second through sixth columns describe five individual synthetic populations, while the final column describes the combined output from all 50 synthetic populations. NH = Non-Hispanic, AI/AN = American Indian/Alaska Native, CNS = central nervous system, DCD = donation after cardiac death

**Figs 2 and 3** provide two visual comparisons between observed and synthetic population characteristics, demonstrating the candidate model's ability to jointly model variation across multiple variables: sex, height, and diagnosis group (Fig 2); and sex, FVC, and diagnosis group (Fig 3). **S1–S40 Figs** provides multiple additional visualizations comparing the characteristics of synthetic and observed populations of donors and candidates. The correlation between the continuous risk factors in the Synthetic candidate populations closely resembled the original

**Table 4. Comparison of observed SRTR and synthetically generated candidate populations.**

| Characteristic | Observed | Synthetic Minimum Count | Synthetic First Quantile Count | Synthetic Median Count | Synthetic Third Quantile Count | Synthetic Maximum Count | 100 Synthetic Runs |
|---|---|---|---|---|---|---|---|
| | N = 18,299* | N = 17,849* | N = 18,186* | N = 18,284* | N = 18,422* | N = 18,741* | N = 1,828,605* |
| **Sex** | | | | | | | |
| **Female** | 42.2% | 42.4% | 41.7% | 43.2% | 42.1% | 42.4% | 42.0% |
| **Male** | 57.8% | 57.6% | 58.3% | 56.8% | 57.9% | 57.6% | 58.0% |
| **Diagnosis Group** | | | | | | | |
| **A** | 23.9% | 23.4% | 24.7% | 23.3% | 23.3% | 24.5% | 23.9% |
| **B** | 5.9% | 5.8% | 5.6% | 6.0% | 6.3% | 5.6% | 5.9% |
| **C** | 7.3% | 7.5% | 7.2% | 7.9% | 7.0% | 7.3% | 7.3% |
| **D** | 62.9% | 63.2% | 62.5% | 62.8% | 63.4% | 62.7% | 62.9% |
| **Race/Ethnicity** | | | | | | | |
| **NH White** | 75.6% | 75.1% | 75.1% | 75.4% | 75.3% | 75.1% | 75.3% |
| **NH Black** | 10.2% | 10.3% | 10.1% | 10.4% | 9.7% | 10.4% | 10.2% |
| **Hispanic** | 10.3% | 10.2% | 10.3% | 10.2% | 10.6% | 10.3% | 10.4% |
| **Asian** | 3.0% | 3.1% | 3.3% | 3.1% | 3.1% | 3.2% | 3.1% |
| **AI/AN** | 0.4% | 0.7% | 0.6% | 0.4% | 0.7% | 0.5% | 0.5% |
| **Pacific Islander** | 0.1% | 0.2% | 0.2% | 0.2% | 0.1% | 0.1% | 0.2% |
| **Multiracial** | 0.3% | 0.3% | 0.4% | 0.3% | 0.4% | 0.4% | 0.4% |
| **Age (years)** | 61 [52, 66]; (29, 72) | 60 [51, 66]; (32, 72) | 60 [51, 66]; (32, 72) | 60 [51, 66]; (31, 72) | 60 [51, 66]; (32, 72) | 60 [51, 66]; (32, 72) | 60 [51, 66]; (31, 72) |
| **Airway Function Index** | -0.11 [-0.56, 0.48]; (-1.02, 1.44) | -0.08 [-0.53, 0.48]; (-1.02, 1.40) | -0.09 [-0.54, 0.45]; (-1.03, 1.36) | -0.10 [-0.56, 0.46]; (-1.04, 1.34) | -0.08 [-0.54, 0.50]; (-1.04, 1.43) | -0.08 [-0.54, 0.48]; (-1.04, 1.38) | -0.09 [-0.55, 0.47]; (-1.04, 1.40) |
| **Oxygen Function Index** | 0.02 [-0.31, 0.36]; (-1.06, 1.16) | 0.03 [-0.31, 0.35]; (-1.05, 1.15) | 0.02 [-0.30, 0.35]; (-1.03, 1.12) | 0.02 [-0.31, 0.34]; (-1.04, 1.12) | 0.03 [-0.31, 0.36]; (-1.04, 1.15) | 0.02 [-0.30, 0.34]; (-1.04, 1.11) | 0.03 [-0.30, 0.35]; (-1.03, 1.13) |
| **Blood Type** | | | | | | | |
| **A** | 37.8% | 37.6% | 37.2% | 38.4% | 37.0% | 37.4% | 37.8% |
| **AB** | 3.7% | 3.7% | 3.6% | 3.5% | 3.6% | 3.8% | 3.7% |
| **B** | 11.7% | 11.7% | 12.2% | 11.8% | 11.8% | 11.5% | 11.7% |
| **O** | 46.8% | 47.1% | 47.0% | 46.4% | 47.7% | 47.3% | 46.8% |
| **Height (cm)** | 170 [163, 178]; (152, 185) | 170 [161, 177]; (152, 185) | 170 [163, 177]; (152, 185) | 169 [161, 177]; (152, 185) | 170 [161, 177]; (152, 185) | 169 [161, 177]; (152, 185) | 170 [161, 177]; (152, 185) |
| **Unknown** | 3 | | | | | | |
| **Weight (kg)** | 75 [62, 86]; (48, 102) | 75 [63, 86]; (48, 101) | 75 [64, 86]; (48, 101) | 75 [63, 86]; (48, 101) | 75 [64, 86]; (48, 101) | 75 [64, 86]; (49, 101) | 75 [64, 86]; (48, 101) |
| **Unknown** | 3 | | | | | | |
| **BMI (km/m²)** | 26.1 [22.4, 29.4]; (18.3, 33.1) | 25.9 [22.6, 29.3]; (18.0, 34.2) | 26.0 [22.6, 29.3]; (18.0, 34.1) | 26.0 [22.7, 29.3]; (18.0, 34.2) | 26.1 [22.7, 29.4]; (18.1, 34.1) | 26.1 [22.8, 29.4]; (18.1, 34.2) | 26.0 [22.7, 29.3]; (18.1, 34.2) |
| **Unknown** | 3 | | | | | | |
| **Diabetes** | 21.0% | 21.1% | 20.1% | 21.1% | 21.1% | 21.7% | 21.2% |
| **Respiratory Support Cluster** | | | | | | | |
| **1** | 14.6% | 15.1% | 13.9% | 14.2% | 15.0% | 14.8% | 14.7% |
| **2** | 66.8% | 66.2% | 67.9% | 66.9% | 66.2% | 67.0% | 66.8% |
| **3** | 15.0% | 15.3% | 14.8% | 15.4% | 15.0% | 14.6% | 15.1% |
| **4** | 3.6% | 3.4% | 3.4% | 3.4% | 3.8% | 3.5% | 3.4% |

(*Continued*)

**Table 4.** (Continued)

| Characteristic | Observed | Synthetic Minimum Count | Synthetic First Quantile Count | Synthetic Median Count | Synthetic Third Quantile Count | Synthetic Maximum Count | 100 Synthetic Runs |
|---|---|---|---|---|---|---|---|
| | N = 18,299* | N = 17,849* | N = 18,186* | N = 18,284* | N = 18,422* | N = 18,741* | N = 1,828,605* |
| **Surgical Type Preference** | | | | | | | |
| **Double** | 59.2% | 60.0% | 58.2% | 59.4% | 59.3% | 57.7% | 59.2% |
| **Either** | 29.2% | 28.3% | 29.7% | 29.2% | 28.8% | 30.6% | 29.2% |
| **Single** | 11.6% | 11.8% | 12.1% | 11.3% | 11.9% | 11.7% | 11.6% |
| **FEV$_1$ (% predicted)** | 39 [24, 56]; (15, 78) | 39 [26, 54]; (15, 76) | 38 [26, 54]; (14, 75) | 39 [26, 54]; (14, 76) | 39 [26, 55]; (14, 77) | 39 [26, 54]; (14, 76) | 39 [26, 54]; (14, 77) |
| **Unknown** | 976 | | | | | | |
| **FVC (% predicted)** | 47 [37, 60]; (25, 82) | 49 [38, 61]; (22, 79) | 49 [38, 61]; (22, 79) | 48 [37, 60]; (22, 79) | 49 [37, 61]; (22, 80) | 49 [38, 61]; (22, 80) | 49 [37, 61]; (22, 80) |
| **Unknown** | 572 | | | | | | |
| **PCO$_2$ (mm Hg)** | 44 [39, 52]; (33, 70) | 45 [39, 52]; (32, 69) | 45 [39, 52]; (33, 70) | 45 [39, 53]; (33, 70) | 45 [39, 52]; (32, 70) | 45 [39, 53]; (32, 70) | 45 [39, 52]; (32, 69) |
| **Unknown** | 515 | | | | | | |
| **P/F ratio** | 281 [224, 343]; (106, 494) | 275 [222, 333]; (129, 481) | 275 [222, 334]; (131, 478) | 274 [222, 333]; (128, 475) | 276 [222, 335]; (132, 481) | 274 [222, 331]; (131, 474) | 275 [222, 333]; (131, 478) |
| **Unknown** | 3,530 | | | | | | |
| **PO$_2$ (mm Hg)** | 69 [57, 89]; (43, 163) | 68 [56, 87]; (44, 149) | 68 [56, 86]; (44, 145) | 68 [56, 86]; (44, 146) | 68 [56, 87]; (45, 148) | 68 [56, 86]; (45, 146) | 68 [56, 86]; (44, 146) |
| **Unknown** | 3,530 | | | | | | |
| **Mean PAP (mm Hg)** | 25 [20, 32]; (14, 51) | 26 [20, 33]; (14, 49) | 25 [20, 32]; (14, 48) | 26 [20, 33]; (14, 49) | 25 [20, 33]; (14, 49) | 25 [20, 32]; (14, 48) | 25 [20, 33]; (14, 48) |
| **Unknown** | 4,052 | | | | | | |
| **Supplemental Oxygen (L/min)** | 4 [3, 7]; (2, 26) | 4 [3, 7]; (2, 26) | 4 [3, 7]; (2, 26) | 4 [3, 7]; (2, 26) | 4 [3, 8]; (2, 26) | 4 [3, 6]; (2, 26) | 4 [3, 7]; (2, 26) |
| **Unknown** | 2 | | | | | | |
| **Oxygen Frequency** | | | | | | | |
| **At Rest** | 85.4% | 84.9% | 85.9% | 85.8% | 84.9% | 85.1% | 85.3% |
| **At Night** | 1.5% | 1.6% | 1.5% | 1.4% | 1.5% | 1.5% | 1.6% |
| **While Exercising** | 10.4% | 10.4% | 10.1% | 9.9% | 10.7% | 10.5% | 10.3% |
| **None** | 2.8% | 3.1% | 2.5% | 2.9% | 2.9% | 2.9% | 2.9% |
| **Ventilator** | | | | | | | |
| **BIPAP** | 7.3% | 7.7% | 7.4% | 7.6% | 7.5% | 7.3% | 7.5% |
| **Mechanical** | 6.4% | 6.1% | 6.1% | 6.4% | 6.3% | 6.5% | 6.2% |
| **None** | 79.7% | 79.4% | 80.1% | 79.4% | 79.6% | 80.1% | 79.7% |
| **CPAP** | 6.6% | 6.8% | 6.4% | 6.5% | 6.7% | 6.1% | 6.6% |
| **Six Minute Walk Distance (feet)** | 816 [480, 1,086]; (0, 1,437) | 806 [518, 1,077]; (0, 1,459) | 793 [517, 1,054]; (0, 1,432) | 788 [500, 1,056]; (0, 1,433) | 794 [520, 1,062]; (0, 1,439) | 788 [515, 1,046]; (0, 1,434) | 795 [515, 1,059]; (0, 1,434) |
| **Unknown** | 140 | | | | | | |
| **Bilirubin (mg/dL)** | 0.50 [0.30, 0.70]; (0.20, 1.20) | 0.46 [0.31, 0.68]; (0.18, 1.18) | 0.47 [0.32, 0.69]; (0.18, 1.19) | 0.47 [0.32, 0.69]; (0.18, 1.19) | 0.47 [0.32, 0.70]; (0.18, 1.22) | 0.46 [0.32, 0.68]; (0.18, 1.17) | 0.47 [0.32, 0.69]; (0.18, 1.19) |

*(Continued)*

**Table 4.** (Continued)

| Characteristic | Observed | Synthetic Minimum Count | Synthetic First Quantile Count | Synthetic Median Count | Synthetic Third Quantile Count | Synthetic Maximum Count | 100 Synthetic Runs |
|---|---|---|---|---|---|---|---|
| | N = 18,299* | N = 17,849* | N = 18,186* | N = 18,284* | N = 18,422* | N = 18,741* | N = 1,828,605* |
| Unknown | 180 | | | | | | |
| Creatinine (mg/dL) | 0.80 [0.68, 0.97]; (0.50, 1.28) | 0.81 [0.66, 0.99]; (0.48, 1.33) | 0.81 [0.66, 0.99]; (0.48, 1.33) | 0.80 [0.65, 0.98]; (0.48, 1.32) | 0.81 [0.66, 0.99]; (0.48, 1.33) | 0.81 [0.66, 0.99]; (0.49, 1.33) | 0.81 [0.65, 0.99]; (0.49, 1.33) |
| Unknown | 26 | | | | | | |
| Systolic PAP (mm Hg) | 39 [32, 50]; (24, 80) | 40 [32, 50]; (23, 74) | 40 [32, 50]; (23, 72) | 40 [32, 51]; (23, 74) | 40 [32, 50]; (23, 74) | 40 [32, 50]; (23, 73) | 40 [32, 50]; (23, 74) |
| Unknown | 3,862 | | | | | | |
| Cardiac Index (L/min/m$^2$) | 2.78 [2.39, 3.26]; (1.87, 4.25) | 2.79 [2.39, 3.27]; (1.90, 4.11) | 2.78 [2.37, 3.26]; (1.88, 4.10) | 2.79 [2.39, 3.27]; (1.91, 4.12) | 2.78 [2.38, 3.26]; (1.89, 4.08) | 2.79 [2.39, 3.25]; (1.91, 4.10) | 2.79 [2.39, 3.27]; (1.90, 4.10) |
| Unknown | 4,309 | | | | | | |
| Central Venous Pressure (mm Hg) | 5.0 [3.0, 8.0]; (0.0, 14.0) | 5.0 [3.0, 8.0]; (1.0, 14.0) | 5.0 [3.0, 8.0]; (1.0, 14.0) | 5.0 [3.0, 8.0]; (1.0, 14.0) | 5.0 [3.0, 8.0]; (1.0, 14.0) | 5.0 [3.0, 8.0]; (1.0, 15.0) | 5.0 [3.0, 8.0]; (1.0, 14.0) |
| Unknown | 4,047 | | | | | | |
| Functional Status | | | | | | | |
| None | 7.8% | 7.4% | 8.4% | 8.2% | 7.5% | 7.8% | 7.8% |
| Some | 82.7% | 83.2% | 82.0% | 82.3% | 82.7% | 83.2% | 82.9% |
| Total | 9.5% | 9.4% | 9.6% | 9.5% | 9.8% | 9.0% | 9.3% |
| ECMO | 3.6% | 3.4% | 3.4% | 3.4% | 3.8% | 3.5% | 3.4% |
| PCO$_2$ increase >15% | 3.2% | 3.1% | 3.0% | 3.5% | 3.6% | 3.2% | 3.3% |
| Unknown | 515 | | | | | | |
| Sarcoidosis Group A | 1.2% | 1.0% | 1.1% | 1.0% | 1.0% | 1.1% | 1.0% |
| Bronchiectasis | 1.8% | 1.9% | 2.2% | 1.7% | 1.6% | 2.0% | 1.9% |
| Lymphangioleio-myomatosis | 0.3% | 0.3% | 0.3% | 0.4% | 0.3% | 0.3% | 0.3% |
| Eisenmenger Syndrome | 0.1% | 0.2% | 0.1% | 0.1% | 0.1% | 0.1% | 0.1% |
| Sarcoidosis Group D | 1.5% | 1.4% | 1.7% | 1.7% | 1.6% | 1.7% | 1.6% |
| Pulmonary Fibrosis (other) | 8.8% | 9.0% | 8.6% | 8.8% | 9.1% | 8.6% | 8.6% |
| Constrictive Bronchiolitis | 0.1% | 0.1% | 0.2% | 0.2% | 0.3% | 0.1% | 0.2% |
| Bronchiolitis Obliterans | 0.8% | 1.0% | 0.7% | 0.7% | 0.8% | 0.7% | 0.8% |

* %; Median [IQR]; (5th percentile, 95th percentile)

The second through sixth columns describe five individual synthetic populations, while the final column describes the combined output from all 50 synthetic populations.

NH = Non-Hispanic; AI/AN = American Indian/Alaska Native; FEV$_1$ = Forced Expiratory Volume in 1 second; FVC = Forced Vital Capacity (% predicted); PCO2 = partial pressure of carbon dioxide; P/F ratio = ratio of partial pressure of arterial oxygen (PaO2) to fraction inspired oxygen (FiPO2); PO2 = partial pressure of oxygen; Mean PAP = mean pulmonary arterial pressure; BIPAP = bi-level positive airway pressure; CPAP = continuous positive airway pressure; Systolic PAP = Systolic pulmonary arterial pressure; CVP = central venous pressure; ECMO = extracorporeal membrane oxygenation. Diagnosis Groups = Group A, obstructive lung disease; Group B, pulmonary vascular disease; Group C, cystic fibrosis and immunodeficiency disorders; and Group D, restrictive lung diseases

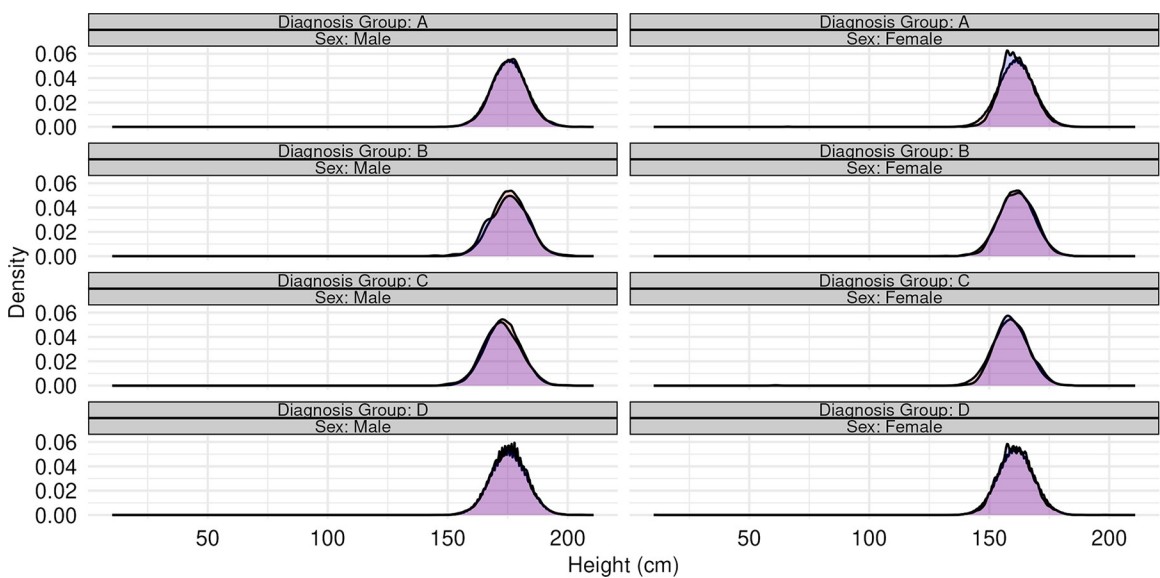

**Fig 2. Ensity plots of candidate height (cm) stratified by sex and diagnosis group.** In each, the original SRTR data is represented with a blue shaded density plot, while the combined output from 100 synthetic populations is represented by a red shaded density plot.

data. (S41 Fig). The joint entropy values for all variables in the donor and candidate synthetic populations were also similar to those in the original data (S42 and S43 Figs).

As an example use case, we generated a synthetic population of candidates based on the projected diagnosis group distribution and group-specific mean ages of transplant candidates from January 1, 2021, through July 15, 2027, and observed the changes in simulated candidate characteristics that would result. **S44 Fig and S1 Table** compare values for all candidate characteristics under the base case scenario versus the scenario incorporating these hypothetical

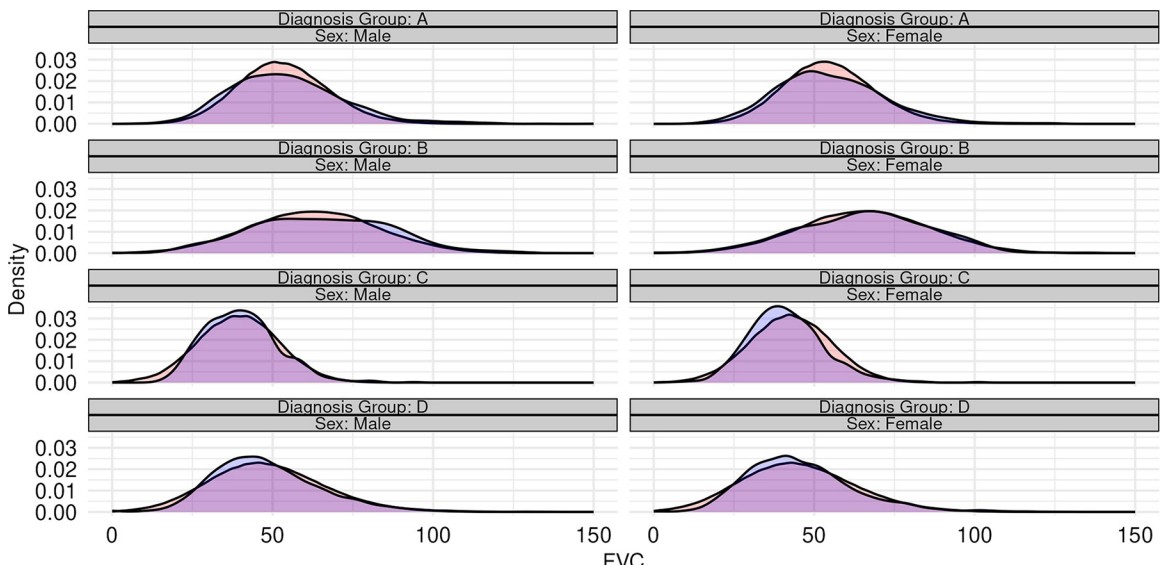

**Fig 3. Density plots of candidate forced vital capacity (FVC, % predicted) stratified by sex and diagnosis group.** In each, the original SRTR data is represented with a blue shaded density plot, while the combined output from 100 synthetic populations is represented by a red shaded density plot.

population trends (100 simulations each). The most notable changes were a decrease in candidates requiring double lung transplants from a median of 59.2% to a median of 57.6%, an increase in median $FEV_1$ (as a percentage of predicted) from 39% to 42%, a decrease in the median proportion of candidates requiring bi-level positive airway pressure (BIPAP) ventilation from 7.5% to 6.3%, and an increase in the median number of patients requiring no ventilatory support from 79.7% to 80.8%.

## Discussion

We have described a method to generate realistic synthetic organ transplant donor and candidate populations using Bayesian models. This approach stands to advance the state of practice in computer simulation of alternative organ transplant policies, currently limited by reliance on data that reflects prior policies and clinical practices. Leveraging synthetic populations untethers developers and decision makers from the use of historical transplant registry data in simulations. Such patient data not only requires substantial confidentiality protections, but it often does not reflect ongoing or anticipated clinical practice advances, policy changes, demographic shifts, or epidemiologic trends. Our validation results showed that, across 100 simulated populations of lung donors and candidates, distributions of donor and candidate characteristics closely matched the actual characteristics observed over corresponding time periods in the U.S. transplant registry. Outputs for a small number of characteristics were slightly biased, but none of them to a clinically meaningful degree. We hypothesize that some of these biases may stem from non-random missingness of SRTR data used to train our models.

We also provided a simple demonstration of how the approach we have described may be used to generate synthetic populations whose clinical characteristics reflect ongoing shifts in the makeup of the candidate pool. Because this method preserves the interdependencies between all candidate characteristics, changing only two upstream population parameter values—diagnosis group-specific mean age and diagnosis group mix in this case—exerts cascading effects on multiple other candidate traits, including clinical traits that would affect a given candidate's priority for transplant or their expected post-transplant survival. In the specific example shown, continuing trends in the rising age of transplant candidates and a continued shift toward higher proportions of Group B (pulmonary vascular disease) and Group D (restrictive lung disease) candidates resulted in a population that tended to have higher FEV1 values, to require less ventilatory support overall (including less use of BIPAP), and to require fewer double lung transplants. The magnitude of the observed change in each individual characteristic was modest, but their combined impact on ultimate simulation outcomes could be substantial.

The purpose of this work is to provide an alternative to the practice of directly re-sampling historical transplant registry data when simulating transplant allocation strategies (whether using the SAM simulation models or others). There are several important implications for avoiding re-sampling. First, our statistical approach offers inherent flexibility to tune parameters in a manner that will generate donor or candidate populations reflective of future states brought about by ongoing or hypothetical changes in demographics, epidemiology, treatment, policy, or transplantation practice patterns. Consider for example the impact of the expanding use of cystic fibrosis transmembrane conductance regulator modulators on individuals with cystic fibrosis, a population whose need for transplant is declining dramatically [38]. A traditional approach to correcting for the fact that earlier SRTR data would not reflect this practice change might be to *under*-sample from these candidates in the SRTR database. However, adjusting for other simultaneous shifts in candidate characteristics stemming from this practice change may be less straightforward. By contrast, it is possible to generate a synthetic population of candidates in which individuals with cystic fibrosis not only represented a smaller

proportion of total candidates but also exhibited less precipitous declines across multiple factors reflecting lung function.

There are numerous structural barriers to patients of minority race/ethnicity with end-stage organ disease becoming transplant candidates [24–26]. A second important implication of applying a synthetic approach to candidate generation is the opportunity to adjust populations in simulations in a manner that acknowledges and attempts to neutralize these disparities. For example, one could adjust model coefficients in a way that changed both the racial/ethnic distribution of candidates and markers of disease severity in a manner that one had reason to believe more accurately reflected the true burden of disease in an underserved racial/ethnic group. Only by simulating allocation in a hypothetical equitable future state—a state reflecting true disease burden—can we design equitable allocation methods and achieve the equity goals long held by the transplant community [24–26].

Important practical implications for simulation modelers also emerge. With the traditional approach of re-sampling donors and candidates from SRTR databases, many candidates will have one or more missing data elements. Sampling from empirically trained probability distributions averts this problem. Finally, foregoing the need to obtain access to the individual-level SRTR U.S. transplant registry data that contains sensitive protected health information can potentially broaden the cadre of researchers simulating alternative transplant policies and scenarios. This open science approach can accelerate the pace of innovation in transplant modeling.

## Leveraging synthetic populations

The work we have described has not been presented as part of a specific allocation simulation model. However, it represents an important component of an emerging open-source transplant simulation framework (article under review) known as the *Computational Open-source Model for Evaluating Transplantation* (COMET). COMET is comprised of interlocking modules, each with standardized inputs and outputs. The synthetic population models presented here can function as modules within the COMET framework. Other COMET modules simulate pre-transplant events, donor-candidate matching, organ acceptance, post-transplant events, etc. This standardization of inputs and outputs for modules in a structured framework allows for targeted innovation within specific simulation substructures.

## Limitations

Building Bayesian network models requires making explicit and formal assumptions about the dependencies between variables. While all presumed associations were confirmed empirically, there may be meaningful dependencies that we neglected to consider. We hope that this work represents a starting point from which we and others will refine and test assumptions about variable dependencies in order to produce increasingly realistic synthetic populations. Our models may also omit relevant variables—comorbid chronic conditions, for example. These models were trained using SRTR data, restricting us to the universe of available registry variables in the U.S. A potential advantage of applying Bayesian networks, however, stems from their modular nature. One could theoretically leverage different data sources to inform subsections of the network, and then test the robustness of the combined network model (incorporating multiple data sources) with new data. This represents an avenue for future research.

## Conclusions

We have introduced a new approach to generating synthetic donor and candidate populations for use in simulations in organ transplantation. The approach uses probabilistic models to

generate populations of individual donor and candidate profiles with realistically co-varying traits. The populations generated by these models closely resembled actual populations of donors and candidates. The application of synthetic populations can potentially advance the use of simulation for studying the effects of demographic, epidemiologic, policy, or practice changes on transplant outcomes–efforts currently hindered by the inability to generate novel populations reflecting existing or future practices.

## Supporting information

**S1 Text. Details of Bayesian Network.**
(DOCX)

**S1 Fig. Box-and-whisker-plot comparing the distribution of sex proportions from 100 synthetic populations to the actual proportions observed in SRTR data (represented by a red dot) covering the period February 19, 2015 and September 1, 2021.** Each box horizontally spans the interquartile range (IQR), while the vertical line inside each box represents the median. The extremes of each whisker mark the observation that is farthest to the left or right of the median while still falling within 1.5 times the IQR. Black dots represent outlying synthetic population counts, defined as greater then +/-1.5 times the IQR from the third and first quartiles respectively.
(TIF)

**S2 Fig. Box-and-whisker-plot comparing the distribution of diagnosis group proportions from 100 synthetic populations to the actual proportions observed in SRTR data (represented by a red dot) covering the period February 19, 2015 and September 1, 2021.**
(TIF)

**S3 Fig. Box-and-whisker-plot comparing the distribution of diagnosis group proportions by sex from 100 synthetic populations to the actual proportions observed in SRTR data (represented by a red dot) covering the period February 19, 2015 and September 1, 2021.**
(TIF)

**S4 Fig. Box-and-whisker-plot comparing the distribution of respiratory support cluster proportions by diagnosis group and sex from 100 synthetic populations to the actual proportions observed in SRTR data (represented by a red dot) covering the period February 19, 2015 and September 1, 2021.**
(TIF)

**S5 Fig. Density plots of candidate airway function index stratified by sex and diagnosis group.** In each, the original SRTR data is represented with a blue shaded density plot, while the combined output from 100 synthetic populations is represented by a red shaded density plot.
(TIF)

**S6 Fig. Density plots of candidate oxygen function index stratified by sex and diagnosis group.** In each, the original SRTR data is represented with a blue shaded density plot, while the combined output from 100 synthetic populations is represented by a red shaded density plot.
(TIF)

**S7 Fig. Density plots of candidate age stratified by diagnosis group.** In each, the original SRTR data is represented with a blue shaded density plot, while the combined output from 100

synthetic populations is represented by a red shaded density plot.
(TIF)

**S8 Fig. Density plots of candidate P/F ratio stratified by sex and diagnosis group.** In each, the original SRTR data is represented with a blue shaded density plot, while the combined output from 100 synthetic populations is represented by a red shaded density plot.
(TIF)

**S9 Fig. Density plots of candidate pO$_2$ stratified by sex and diagnosis group.** In each, the original SRTR data is represented with a blue shaded density plot, while the combined output from 100 synthetic populations is represented by a red shaded density plot.
(TIF)

**S10 Fig. Density plots of candidate mean pulmonary artery pressure (PAP) stratified by sex and diagnosis group.** In each, the original SRTR data is represented with a blue shaded density plot, while the combined output from 100 synthetic populations is represented by a red shaded density plot.
(TIF)

**S11 Fig. Density plots of candidate FEV$_1$ stratified by sex and diagnosis group.** In each, the original SRTR data is represented with a blue shaded density plot, while the combined output from 100 synthetic populations is represented by a red shaded density plot.
(TIF)

**S12 Fig. Density plots of candidate pCO$_2$ stratified by sex and diagnosis group.** In each, the original SRTR data is represented with a blue shaded density plot, while the combined output from 100 synthetic populations is represented by a red shaded density plot.
(TIF)

**S13 Fig. Box-and-whisker-plot comparing the distribution of ventilation type proportions from 100 synthetic populations to the actual proportions observed in SRTR data (represented by a red dot) covering the period February 19, 2015 and September 1, 2021.**
(TIF)

**S14 Fig. Bar plot comparing the distribution of supplemental oxygen (L/min) stratified by diagnosis group from 100 synthetic populations to the actual proportions observed in SRTR data (represented by a red dot) covering the period February 19, 2015 and September 1, 2021.** In each, the original SRTR data is represented with a blue shaded bars, while the combined output from 100 synthetic populations is represented by a red shaded bars.
(TIF)

**S15 Fig. Density plots of candidate six minute walk distance (ft) stratified by respiratory support cluster and diagnosis group.** In each, the original SRTR data is represented with a blue shaded density plot, while the combined output from 100 synthetic populations is represented by a red shaded density plot.
(TIF)

**S16 Fig. Box-and-whisker-plot comparing the distribution of oxygen frequency proportions by respiratory support cluster from 100 synthetic populations to the actual proportions observed in SRTR data (represented by a red dot) covering the period February 19, 2015 and September 1, 2021.**
(TIF)

**S17 Fig. Box-and-whisker-plot comparing the distribution of race and ethnicity proportions from 100 synthetic populations to the actual proportions observed in SRTR data (represented by a red dot) covering the period February 19, 2015 and September 1, 2021.**
(TIF)

**S18 Fig. Box-and-whisker-plot comparing the distribution of blood type proportions by race/ethnicity from 100 synthetic populations to the actual proportions observed in SRTR data (represented by a red dot) covering the period February 19, 2015 and September 1, 2021.**
(TIF)

**S19 Fig. Density plots of candidate weight (kg) stratified by sex and diagnosis group.** In each, the original SRTR data is represented with a blue shaded density plot, while the combined output from 100 synthetic populations is represented by a red shaded density plot.
(TIF)

**S20 Fig. Density plots of candidate BMI stratified by sex and diagnosis group.** In each, the original SRTR data is represented with a blue shaded density plot, while the combined output from 100 synthetic populations is represented by a red shaded density plot.
(TIF)

**S21 Fig. Density plots of candidate bilirubin stratified by diagnosis group.** In each, the original SRTR data is represented with a blue shaded density plot, while the combined output from 100 synthetic populations is represented by a red shaded density plot.
(TIF)

**S22 Fig. Density plots of candidate creatinine stratified by diagnosis group.** In each, the original SRTR data is represented with a blue shaded density plot, while the combined output from 100 synthetic populations is represented by a red shaded density plot.
(TIF)

**S23 Fig. Density plots of candidate cardiac index stratified by diagnosis group.** In each, the original SRTR data is represented with a blue shaded density plot, while the combined output from 100 synthetic populations is represented by a red shaded density plot.
(TIF)

**S24 Fig. Density plots of candidate systolic pulmonary artery pressure (PAP) stratified by diagnosis group.** In each, the original SRTR data is represented with a blue shaded density plot, while the combined output from 100 synthetic populations is represented by a red shaded density plot.
(TIF)

**S25 Fig. Box-and-whisker-plot comparing the distribution of specific end-stage lung diagnoses from 100 synthetic populations to the actual proportions observed in SRTR data (represented by a red dot) covering the period February 19, 2015 and September 1, 2021.**
(TIF)

**S26 Fig. Box-and-whisker-plot comparing the distribution of $pCO_2$ threshold increase of 15% from 100 synthetic populations to the actual proportions observed in SRTR data (represented by a red dot) covering the period February 19, 2015 and September 1, 2021.**
(TIF)

**S27 Fig. Box-and-whisker-plot comparing the distribution of lung surgery type stratified by diagnosis group from 100 synthetic populations to the actual proportions observed in**

SRTR data (represented by a red dot) covering the period February 19, 2015 and September 1, 2021.
(TIF)

**S28 Fig. Density plots of candidate central venous pressure stratified by diagnosis group.** In each, the original SRTR data is represented with a blue shaded density plot, while the combined output from 100 synthetic populations is represented by a red shaded density plot.
(TIF)

**S29 Fig. Box-and-whisker-plot comparing the diabetes stratified by race/ethnicity and sex from 100 synthetic populations to the actual proportions observed in SRTR data (represented by a red dot) covering the period February 19, 2015 and September 1, 2021.**
(TIF)

**S30 Fig. Scatter plot of donor count by donor hospital.** Each red dot represents a donor hospital. The x-axis is the observed count in SRTR from January 1st, 2015 to June 30th 2021. The y-axis is the median simulated count. The vertical bars with each red dot represent the simulated IQR for each donor hospital.
(TIF)

**S31 Fig. Box-and-whisker-plot comparing the distribution of donor sex from 100 synthetic populations to the actual proportions observed in SRTR data (represented by a red dot) covering the period January 1, 2015 and June 30, 2021.**
(TIF)

**S32 Fig. Density plots of donor height (cm) stratified.** In each, the original SRTR data is represented with a blue shaded density plot, while the combined output from 100 synthetic populations is represented by a red shaded density plot.
(TIF)

**S33 Fig. Density plots of donor height (cm) stratified by sex.** In each, the original SRTR data is represented with a blue shaded density plot, while the combined output from 100 synthetic populations is represented by a red shaded density plot.
(TIF)

**S34 Fig. Box-and-whisker-plot comparing the distribution of donor race/ethnicity from 100 synthetic populations to the actual proportions observed in SRTR data (represented by a red dot) covering the period January 1, 2015 and June 30, 2021.**
(TIF)

**S35 Fig. Box-and-whisker-plot comparing the distribution of donor blood type stratified by race/ethnicity from 100 synthetic populations to the actual proportions observed in SRTR data (represented by a red dot) covering the period January 1, 2015 and June 30, 2021.**
(TIF)

**S36 Fig. Box-and-whisker-plot comparing the distribution of donor cause of death from 100 synthetic populations to the actual proportions observed in SRTR data (represented by a red dot) covering the period January 1, 2015 and June 30, 2021.**
(TIF)

**S37 Fig. Density plots of donor age stratified by cause of death.** In each, the original SRTR data is represented with a blue shaded density plot, while the combined output from 100

synthetic populations is represented by a red shaded density plot.
(TIF)

**S38 Fig. Box-and-whisker-plot comparing the distribution of donor smoking history (>20 pack years) from 100 synthetic populations to the actual proportions observed in SRTR data (represented by a red dot) covering the period January 1, 2015 and June 30, 2021.**
(TIF)

**S39 Fig. Box-and-whisker-plot comparing the distribution of donor organs available from 100 synthetic populations to the actual proportions observed in SRTR data (represented by a red dot) covering the period January 1, 2015 and June 30, 2021.**
(TIF)

**S40 Fig. Box-and-whisker-plot comparing the distribution of donor DCD status from 100 synthetic populations to the actual proportions observed in SRTR data (represented by a red dot) covering the period January 1, 2015 and June 30, 2021.**
(TIF)

**S41 Fig. Heatmap displaying correlation coefficients of continuous candidate characteristics.** The panel on the left represents correlations from 100 synthetic candidate populations, while the panel on the right summarizes the corresponding correlations from the original candidate cohort from February 19, 2015 through September 1, 2021. Pairwise complete cases were used for the original population in the event of missing data. One outlier of BMI was removed the correlation calculations involving BMI but was included in the calculations for the other variables.
(TIF)

**S42 Fig. Heatmap displaying joint entropy values for donor characteristics.** The panel on the left represents the synthetically generated candidates across 100 populations, while the panel on the right represents the original candidate cohort from January 1, 2015 to June 30, 2021. Pairwise complete cases were used for the original population in the event of missing data. To calculate joint entropy with continuous variables, all continuous variables were best split into 4 groups of equal size based on the original data, and these same numerical splits were applied to the synthetic data.
(TIF)

**S43 Fig. Heatmap displaying joint entropy values for candidate characteristics.** The panel on the left represents the synthetically generated candidates across 100 populations, while the panel on the right represents the original candidate cohort from February 19, 2015 and September 1, 2021. Pairwise complete cases were used for the original population in the event of missing data. To calculate joint entropy with continuous variables, all continuous variables were split into 4 groups of equal size based on the original data, and these same numerical splits were applied to the synthetic data.
(TIF)

**S44 Fig.** Panels A and B are violin plots comparing the distribution of candidate surgery types and ventilator status respectively from 100 synthetic populations with the original parameters and the diagnosis group and age extrapolations. Panels C and D are density plots comparing the distribution of FEV1 and FVC respectively for 100 synthetic populations with the original parameters and the diagnosis group and age extrapolations.
(TIF)

**S1 Table. Comparison of candidate population composition under two scenarios.**
(DOCX)

## Acknowledgments

The data reported here have been supplied by the Hennepin Healthcare Research Institute (HHRI) as the contractor for the Scientific Registry of Transplant Recipients (SRTR). The interpretation and reporting of these data are the responsibility of the author(s) and in no way should be seen as an official policy of or interpretation by the SRTR or the U.S. Government. The authors thank Cleveland Clinic colleague Jodi Bell, MA for manuscript editing.

## Author Contributions

**Conceptualization:** Paul R. Gunsalus, Johnie Rose, Carli J. Lehr, Maryam Valapour, Jarrod E. Dalton.

**Formal analysis:** Paul R. Gunsalus, Johnie Rose, Jarrod E. Dalton.

**Methodology:** Paul R. Gunsalus, Johnie Rose, Jarrod E. Dalton.

**Writing – original draft:** Paul R. Gunsalus, Johnie Rose.

**Writing – review & editing:** Carli J. Lehr, Maryam Valapour, Jarrod E. Dalton.

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
