## [Decision Letter · Decision Letter 0]

15 Oct 2023

PONE-D-23-30778Creating synthetic populations in transplantation: A Bayesian approach enabling simulation without registry re-samplingPLOS ONE

Dear Dr. Rose,

Thank you for submitting your manuscript to PLOS ONE. After careful consideration, we feel that it has merit but does not fully meet PLOS ONE’s publication criteria as it currently stands. Therefore, we invite you to submit a revised version of the manuscript that addresses the points raised during the review process.

We look forward to receiving your revised manuscript. PLEASE NOTE THAT SOME REVIEWER COMMENTS ARE SUBMITTED AS PART OF AN ATTACHMENT. PLEASE INCLUDE THEM IN YOUR RESPONSES.

Kind regards,

Jaimin R. Trivedi, MBBS, MPH

Academic Editor

PLOS ONE

[This work was conducted under the auspices of the Hennepin Healthcare Research Institute (HHRI), contractor for the Scientific Registry of Transplant Recipients (SRTR), under contract no. 75R60220C00011 (US Department of Health and Human Services, Health Resources and Services Administration, Health Systems Bureau, Division of Transplantation). The US Government (and others acting on its behalf) retains a paid-up, nonexclusive, irrevocable, worldwide license for all works produced under the SRTR contract, and to reproduce them, prepare derivative works, distribute copies to the public, and perform publicly and display publicly, by or on behalf of the Government. The data reported here have been supplied by HHRI as the contractor for SRTR. The interpretation and reporting of these data are the responsibility of the author(s) and in no way should be seen as an official policy of or interpretation by the SRTR or the U.S. Government. The authors thank Cleveland Clinic colleague Jodi Bell, MA for manuscript editing.]

 [This project was funded by the Heart, Lung, and Blood Institute of the National Institutes of Health (NHLBI) R01HL153175. Dr. Lehr is supported by NHLBI K08HL159236. The content is solely the responsibility of the authors and does not necessarily represent the official views of the National Institutes of Health. The funding sources had no role in the design or conduct of the study; collection, management, analyses, or interpretation of the data; preparation, review, or approval of the manuscript; or decision to submit the manuscript for publication. The funding sources did not provide input or contribute to the analysis or conclusions of this manuscript.]

Reviewers' comments:

Reviewer's Responses to Questions

**Comments to the Author**

1. Is the manuscript technically sound, and do the data support the conclusions?

Reviewer #1: Yes

Reviewer #2: Yes

Reviewer #3: Partly

2. Has the statistical analysis been performed appropriately and rigorously? 

Reviewer #1: I Don't Know

Reviewer #2: Yes

Reviewer #3: Yes

3. Have the authors made all data underlying the findings in their manuscript fully available?

Reviewer #1: Yes

Reviewer #2: No

Reviewer #3: Yes

4. Is the manuscript presented in an intelligible fashion and written in standard English?

Reviewer #1: Yes

Reviewer #2: Yes

Reviewer #3: Yes

5. Review Comments to the Author

Reviewer #1: This manuscript - "Creating synthetic populations in transplantation: A Bayesian approach enabling simulation without registry re-sampling" by Johnie Rose et al - describes a method for generating simulated lung transplant donor and recipient populations, with the goal of being able to use these simulations to predict the impact of demographic shifts or policy changes. The concept is well described, and the manuscript well written. I have a few questions and comments:

1) As with most models, picking the appropriate variables, and determining the associations between them, determines the utility of the model. On the donor side of the model, was there any consideration to including the paO2/FiO2 ratio for the donor lungs as a variable? This is the single most important clinical variable regarding the likelihood of the lungs being accepted, as well a critical factor in the success of the transplant. As such, knowing how the model expects this parameter to change with shift in demographics would be very important.

2) Similarly, ex vivo lung perfusion has been an increasing factor in the utilization of donors, especially with the increase in the use of DCD for lung procurement. Knowing if changes will lead to an increase in the need for ex vivo lung perfusion would also be of great clinical import, since not all centers have access to this technology.

3) I appreciate that the entirety of the network models would be difficult to present, but the fact that you felt the need to develop three indices/clusters to keep certain variables grouped leads one to wonder about the organization of the rest of the model. Can you give some additional description about how the network was organized? You state "presumed dependencies between variables were based on known clinical and physiologic relationships" - which leaves quite a bit to the imagination.

4) Some of your answer to question #3 will no doubt involve pointing to your simulation data, showing that you accurately simulated the existing data, which is a strong point your favor. I do find some of the discrepancies to be interesting - for example, why do you think the simulated population is shorter than the actual population? You explain away some of the other discrepancies by saying there was incomplete data, but that should not be an issue for height, as neither the donors nor recipients would be able to match without a height entered into the system.

5) For your example use case, if I understand it correctly, you simulated a population several years into the future, presuming that there would be a demographic shift away from cystic fibrosis patients - the scope of this shift was based on extrapolating the current demographic shifts? I find it interesting that you also picked a change in age as the other shift for this use case, as shifting away from cystic fibrosis patients (who are generally much younger) will also shift the average age of the transplanted patient towards older. Can you tell which of your changes led to what impacts downstream? And are you able to separate influences of the two demographic shifts on the overall age - and how that age shift impacts the other variables?

6) While achieving equity in lung transplant from a racial/ethinic perspective is obviously a laudable goal, I struggle somewhat to understand how this modeling system would facilitate this process. You state that you could "adjust populations in simulations in a manner that acknowledges and attempts to neutralize these disparities" - what does this mean, exactly? How would you adjust the population? If you just increase, for example, the number of African-American recipients, how do you know that the model is going to respond correctly, given that the model is based, and tested, on a population that is predominantly caucasian?

Reviewer #2: The manuscript addresses an interesting topic. Synthetic control methods are widely used in the medical statistics and economics literature. Novel methods are always welcomed, and the proposal is promising. Numerical examples are given to show the goodness of performance of the proposed method. Some comments follow.

1. I feel that there is room left to improve the description of the methodological part. The implementation of the method, as well as statistical inference should be described more in depth. Details about the hierarchical Bayesian Poisson model are currently swept under the carpet. Info about the priors and other computational details should be included to ensure the reproducibility of the results. From a methodological perspective, I am wondering if large sample inferential techniques, known to be not well suited to comparative case studies when the number of units in the comparison group is small, are still valid; in general, uncertainty measures should be included and discussed. Similarly, more details about computational times, and the employed algorithm should be introduced.

2. A comparison with other well-established approaches is currently missing. To really appreciate the advantages of the proposal with respect to existing methods, comparisons are mandatory. I am not sure that it is straightforward to compare your approach with that in Abadie A, Diamond A, Hainmueller J (2010). “Synthetic Control Methods for Comparative Case Studies: Estimating the Effect of California’s Tobacco Control Program.” Journal of

the American Statistical Association, 105(490), 493–505. (implemented in the Synth R package), but this is of course a good reference to start from.

3. Model validation is crucial. The impact of atypical values, missingness, etc should be discussed and solutions to these potential issues discussed. Similarly, did you check for the reliability of the Poisson assumption? Why did not you consider e.g. a Negative Binomial or another count distribution? Is there any difficulties in moving out from the Poisson case?

Reviewer #3: See attached file.

6. PLOS authors have the option to publish the peer review history of their article (what does this mean?). If published, this will include your full peer review and any attached files.

Reviewer #1: No

Reviewer #2: No

Reviewer #3: No

---

## [Decision Letter · Decision Letter 1]

20 Dec 2023

Creating synthetic populations in transplantation: A Bayesian approach enabling simulation without registry re-sampling

PONE-D-23-30778R1

Dear Dr. Rose,

We’re pleased to inform you that your manuscript has been judged scientifically suitable for publication and will be formally accepted for publication once it meets all outstanding technical requirements.

Kind regards,

Jaimin R. Trivedi, MBBS, MPH

Academic Editor

PLOS ONE

Additional Editor Comments (optional):

Reviewers' comments:

Reviewer's Responses to Questions

**Comments to the Author**

1. If the authors have adequately addressed your comments raised in a previous round of review and you feel that this manuscript is now acceptable for publication, you may indicate that here to bypass the “Comments to the Author” section, enter your conflict of interest statement in the “Confidential to Editor” section, and submit your "Accept" recommendation.

Reviewer #1: All comments have been addressed

Reviewer #3: All comments have been addressed

2. Is the manuscript technically sound, and do the data support the conclusions?

Reviewer #1: Yes

Reviewer #3: Yes

3. Has the statistical analysis been performed appropriately and rigorously? 

Reviewer #1: Yes

Reviewer #3: Yes

4. Have the authors made all data underlying the findings in their manuscript fully available?

Reviewer #1: Yes

Reviewer #3: Yes

5. Is the manuscript presented in an intelligible fashion and written in standard English?

Reviewer #1: Yes

Reviewer #3: Yes

6. Review Comments to the Author

Reviewer #1: (No Response)

Reviewer #3: The majority of my concerns from the previous review have been satisfactorily addressed. While a comparison using an alternatively constructed simulated data would be useful, it may not be necessary.

7. PLOS authors have the option to publish the peer review history of their article (what does this mean?). If published, this will include your full peer review and any attached files.

Reviewer #1: No

Reviewer #3: No

---

## [Editor Report · Acceptance letter]

11 Mar 2024

PONE-D-23-30778R1 

PLOS ONE

Dear Dr. Rose, 

I'm pleased to inform you that your manuscript has been deemed suitable for publication in PLOS ONE. Congratulations! Your manuscript is now being handed over to our production team.

Kind regards, 

on behalf of

Dr. Jaimin R. Trivedi 

Academic Editor

PLOS ONE